# Tiered Pruning for Efficient Differentiable Inference-Aware Neural Architecture Search

## Abstract

We propose three novel pruning techniques to improve the cost and results of Inference-Aware Differentiable Neural Architecture Search (DNAS). First, we introduce **Prunode**, a stochastic bi-path building block for DNAS, which can search over inner hidden dimensions with $\mathcal{O}(1)$ memory and compute complexity. Second, we present an algorithm for pruning blocks within a stochastic layer of the SuperNet during the search. Third, we describe a novel technique for pruning unnecessary stochastic layers during the search. The optimized models resulting from the search are called PRUNET and establishes a new state-of-the-art Pareto frontier for NVIDIA V100 in terms of inference latency for ImageNet Top-1 image classification accuracy. PRUNET as a backbone also outperforms GPUNet and EfficientNet on the COCO object detection task on inference latency relative to mean Average Precision (mAP).

## 1 Introduction

Neural Architecture Search (NAS) is a well-established technique in Deep Learning (DL); conceptually it is comprised of a search space of permissible neural architectures, a search strategy to sample architectures from this space, and an evaluation method to assess the performance of the selected architectures. Because of practical reasons, Inference-Aware Neural Architecture Search is the cornerstone of the modern Deep Learning application deployment process. Wang et al. (2022); Wu et al. (2019); Yang et al. (2018) use NAS to directly optimize inference specific metrics (e.g., latency) on targeted devices instead of limiting the model's `FLOPs` or other proxies. Inference-Aware NAS streamlines the development-to-deployment process. New concepts in NAS succeed with the ever-

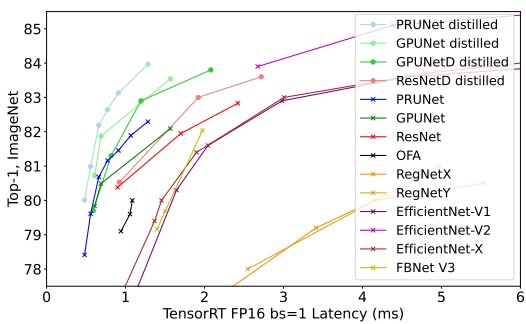

Figure 1: PRUNET establishes a new state-of-the-art Pareto frontier in terms of inference latency for ImageNet Top-1 image classification accuracy.

growing search space, increasing the dimensionality and complexity of the problem. Balancing the search-cost and quality of the search hence is essential for employing NAS in practice.

Traditional NAS methods require evaluating many candidate networks to find optimized ones with respect to the desired metric. This approach can be successfully applied to simple problems like CIFAR-10 Krizhevsky et al. (2010), but for more demanding problems, these methods may turn out to be computationally prohibitive. To minimize this computational cost, recent research has focused on partial training Falkner et al. (2018); Li et al. (2020a); Luo et al. (2018), performing network morphism Cai et al. (2018a); Jin et al. (2019); Molchanov et al. (2021) instead of training from scratch, or training many candidates at the same time by sharing the weights Pham et al. (2018). These approaches can save computational time, but their reliability is questionable Bender et al. (2018); Xiang et al. (2021); Yu et al. (2021); Liang et al. (2019); Chen et al. (2019); Zela et al. (2019), i.e., the final result can still be improved. In our experiments, we focus on a search space based on a state-of-the-art network to showcase the value of our methodology. We aim to revise the

weight-sharing approach to save resources and improve the method's reliability by introducing novel pruning techniques described below.

**Prunode: pruning internal structure of the block**    In the classical SuperNet approach, search space is defined by the initial SuperNet architecture. That means GPU memory capacity significantly limits search space size. In many practical use cases, one would limit themselves to just a few candidates per block. For example, FBNet Wu et al. (2019) defined nine candidates: skip connection and 8 Inverted Residual Blocks (IRB) with expansion, kernel, group $\in$ $\{(1,3,1),(1,3,2),(1,5,1),(1,5,2),(3,3,1),(3,5,1),(6,3,1),(6,5,1)\}$. In particular, one can see that the expansion parameter was limited to only three options: 1, 3, and 6, while more options could be considered - not only larger values but also denser sampling using non-integer values. Each additional parameter increases memory and compute costs, while only promising ones can improve the search. Selecting right parameters for a search space requires domain knowledge. To solve this problem, we introduce a special multi-block called Prunode, which optimizes the value of parameters, such as the expansion parameter in the IRB block. The computation and memory cost of Prunode is equal to the cost of calculating two candidates. Essentially, the Prunode in each iteration emulates just two candidates, each with a different number of channels in the internal structure. These candidates are modified based on the current architecture weights. The procedure encourages convergence towards an optimal number of channels.

**Pruning blocks within a stochastic layer**    In the classical SuperNet approach, all candidates are trained together throughout the search procedure, but ultimately, one or a few candidates are sampled as a result of the search. Hence, large amounts of resources are devoted to training blocks that are ultimately unused. Moreover, since they are trained together, results can be biased due to co-adaptation among operations Bender et al. (2018). We introduce progressive SuperNet pruning based on trained architecture weights to address this problem. This methodology removes blocks from the search space when the likelihood of the block being sampled is below a linearly changing threshold. Reduction of the size of the search space saves unnecessary computation cost and reduces the co-adoption among operations.

**Pruning unnecessary stochastic layers**    By default, layer-wise SuperNet approaches force all networks that can be sampled from the search space to have the same number of layers, which is very limiting. That is why it is common to use a skip connection as an alternative to residual blocks in order to mimic shallower networks. Unfortunately, skip connections blocks' output provides biased information when averaged with the outputs of other blocks. Because of this, SuperNet may tend to sample networks that are shallower than optimal. To solve this problem, we provide a novel method for skipping whole layers in a SuperNet. It introduces the skip connection to the SuperNet during the procedure. Because of that, the skip connection is not present in the search space at the beginning of the search. Once the skip connection is added to the search space, the outputs of the remaining blocks are multiplied by coefficients.

## 2    RELATED WORKS

In NAS literature, a widely known SuperNet approach Liu et al. (2018b); Wu et al. (2019) constructs a stochastic network. At the end of the architecture search, the final non-stochastic network is sampled from the SuperNet using differentiable architecture parameters. The PRUNET algorithm utilizes this scheme – it is based on the weight-sharing design Cai et al. (2019); Wan et al. (2020) and it relies on the Gumbel-Softmax distribution Jang et al. (2016).

The PRUNET algorithm is agnostic to one-shot NAS Liu et al. (2018b); Pham et al. (2018); Wu et al. (2019) where only one SuperNet is trained or few-shot NAS Zhao et al. (2021) where multiple SuperNets were trained to improve the accuracy. In this work, we evaluate our method on search space based on the state-of-the-art GPUNet model Wang et al. (2022). We follow its structure including the number of channels, the number of layers, and the basic block types.

Other methods that incorporate NAS Dai et al. (2019); Dong et al. (2018); Tan et al. (2019) but remain computationally expensive. Differentiable NAS  Cai et al. (2018b); Vahdat et al. (2020); Wu et al. (2019) significantly reduces the training cost. MobileNets Howard et al. (2017); Sandler et al. (2018) started to discuss the importance of model size and latency on embedded systems while

inference-aware pruning Cai et al. (2018b); Howard et al. (2017); Sandler et al. (2018); Wang et al. (2022) is usually focused on compressing fixed architectures Chen et al. (2018); Shen et al. (2021); Yang et al. (2018). Molchanov et al. (2021) further combines layer-wise knowledge-distillation (KD) with NAS to speedup the validation.

DNAS and weight-sharing approaches have two major opposing problems, which we address in this article: memory costs that bound the search space and co-adaptation among operation problem for large search spaces. We also compare our methodology to other similar SuperNet pruning techniques.

**Memory cost**   The best-known work on the memory cost problem is FBNetV2 Wan et al. (2020). The authors use different kinds of masking to mimic many candidates with $\mathcal{O}(1)$ memory and compute complexity. Unfortunately, masking removes information from the feature maps, which results in bias after averaging with informative feature maps (see Section 3.2.3). Therefore, it may lead to suboptimal sampling. FBNetV2 can extend the search space even $10^{14}$ times through this application. Unfortunately, such a large expansion may increase the impact of the co-adaptation among operations problem. Moreover, it can render block benchmarking computationally expensive.

**Co-adaptation among operations**   The weight-sharing approach forces all the operations to be used in multiple contexts, losing the ability to evaluate them individually. This problem, named co-adaptation among operations, posterior fading, or curse of skip connect was noticed by many researchers Bender et al. (2018); Li et al. (2020b); Zhao et al. (2021); Ding et al. (2022). Progressive search space shrinking Li et al. (2020b); Liu et al. (2018a) in different forms was applied to speed up and improve Neural Architecture Search.

**SuperNet pruning**   Ideas similar to our "Pruning blocks within a stochastic layer" modification (see Section 3.2.2) has already been considered in literature. Xia et al. (2022) removes blocks from the search space which are rarely chosen. In Ci et al. (2021), the authors efficiently search over a large search space with multiple SuperNet trainings. In contrast to both methods, our approach requires a single training of a SuperNet, which makes the procedure much faster. Ding et al. (2022) uses DARTs Liu et al. (2018b) to prune operations in a cell. It assesses the importance of each operation and all operations but the two strongest are removed at the end of the training. In contrast, our methodology removes blocks from the SuperNet during the training, reducing the training time.

## 3   METHODOLOGY

Our method can be applied both to layer-wise and cell-wise searches, cf. Bender et al. (2018); Liu et al. (2018b); Mei et al. (2019); Xie et al. (2018). Cell-wise search aims to find a cell (building block), which is then repeated to build the whole network. However, as stated in Lin et al. (2020), different building blocks might be optimal in different network parts. That is why we focus on layer-wise search, i.e., each layer can use a different building block.

The SuperNet is assumed to have a layered structure. Each layer is built from high-level blocks, e.g., IRB in computer vision or self-attention blocks in natural language processing. Only one block in each layer is selected in the final architecture. Residual layers can be replaced by skip connections during the search, effectively making the network shallower.

### 3.1   SEARCH SPACE

Since layer-wise SuperNets must have a predetermined number of output channels for each layer, selecting the right numbers is critical. Thus, it is worth getting inspired by a good baseline model and defining the search space such that models similar to the baseline model can be sampled from the SuperNet. With a search space defined this way, we can expect to find models uniformly better than the baseline or models with different properties, e.g., optimized accuracy for target latency or optimized latency for a target accuracy. In our main experiments, we chose GPUNet-1 Wang et al. (2022) as a baseline network. In Table 1 we present the structure of our SuperNet defining the search space. Since we sample all possible expansions with channel granularity set to 32 (i.e., the number of channels is forced to be a multiple of 32) and skip connections are considered for all residual layers, our search space is covering $3^2 * 24 * 65 * 64 * 97^2 * 96 * 161 * 160 * 289^4 * 288 * 449^4 \approx 1.7e39$

Table 1: PRUNET search space inspired by GPUNet-1

| Stage | Type | Stride | Kernel | # Layers | Activation | Expansion | Filters | SE |
|-------|------|--------|--------|----------|------------|-----------|---------|-----|
| 0 | Conv | 2 | 3 | 1 | Swish | | 24 | |
| 1 | Conv | 1 | {3,5} | 2 | RELU | | 24 | |
| 2 | Fused-IRB | 2 | {3,5} | 3 | Swish | $(0, 8]$ | 64 | {0,1} |
| 3 | Fused-IRB | 2 | {3,5} | 3 | Swish | $(0, 8]$ | 96 | {0,1} |
| 4 | IRB | 2 | {3,5} | 2 | Swish | $(0, 8]$ | 160 | {0,1} |
| 5 | IRB | 1 | {3,5} | 5 | RELU | $(0, 8]$ | 288 | {0,1} |
| 6 | IRB | 2 | {3,5} | 6 | RELU | $(0, 8]$ | 448 | {0,1} |
| 7 | Conv + Pool + FC | 1 | 1 | 1 | RELU | | 1280 | |

candidates. For comparison, similar FBNet SuperNet, which would consume the same amount of GPU memory, would cover only $2^2 * 8^{17} \approx 9e15$ candidates. So, in this case, our methodology can cover $\approx 1.9e23$ times more options with the same GPU memory consumption.

## 3.2 SEARCH METHOD

Our search algorithm is similar to Wu et al. (2019). We focus on multi-objective optimization

$$\min_{\psi} \min_{\theta} L(\psi, \theta), \tag{1}$$

where $\psi$ are the network weights and $\theta$ are the architecture weights. The goal is to minimize the following inference-aware loss function $L$

$$L(\psi, \theta) = \text{CE}(\psi, \theta) + \alpha \log(\text{LAT}(\theta))^{\beta}, \tag{2}$$

where $\text{CE}(\psi, \theta)$ is the standard cross-entropy loss, and $\text{LAT}(\theta)$ is the latency of the network. Coefficient $\alpha$ defines the trade-off between accuracy and latency. Higher $\alpha$ results in finding networks with lower latency; lower $\alpha$ results in finding networks with higher accuracy. Coefficient $\beta$ scales the magnitude of the latency. The loss function is inference-aware, meaning we optimize the latency of the networks for specific hardware.

We train the SuperNet using continuous relaxation. Output of $l$-th layer is equal to

$$x_{l+1} = \sum_i a_{l,i} \cdot B_{l,i}(x_l), \tag{3}$$

where $x_l$ is the output of the layer $(l-1)$, $B_{l,i}(x_l)$ represents the output of the $i$-th block of the $l$-th layer. Every block in a layer is assumed to have the same input and output tensor shapes. The coefficients $a_{l,i}$ come from the Gumbel-Softmax distributionJang et al. (2016)

$$a_{l,i} = \text{Gumbel-Softmax}(\theta_{l,i}|\theta_l) = \frac{exp[(\theta_{l,i} + g_{l,i})/\tau]}{\sum_j exp[(\theta_{l,i} + g_{l,i})/\tau]}, \tag{4}$$

where $\theta_{l,i}$ is the architecture weight of the block. $g_{l,i}$ is sampled from Gumbel$(0, 1)$. Parameter $\tau$ controls the temperature of the Gumbel-Softmax function. In contrast to Wu et al. (2019), we set $\tau$ to have a constant value throughout the training. Architecture weights $\theta$ are differentiable and trained using gradient descent alongside $\psi$ weights.

The total latency $\text{LAT}(\theta)$ is the sum of latencies of all layers in the SuperNet. Similarly, the latency of a layer is a sum of latencies of its blocks weighted by Gumbel-Softmax coefficients

$$\text{LAT}(\theta) = \sum_l \sum_i a_{l,i} \cdot \text{LAT}(B_{l,i}). \tag{5}$$

Our method allows choosing the target device for inference. On chosen hardware, we pre-compute and store in the lookup table the latency $\text{LAT}(B_{l,i})$ for every permissible block, the same way as in Wu et al. (2019). Then the search does not need to compute any further latencies and can be run on arbitrary hardware (e.g., for embedded target hardware used in autonomous vehicles, one could still train the network on a supercomputer).

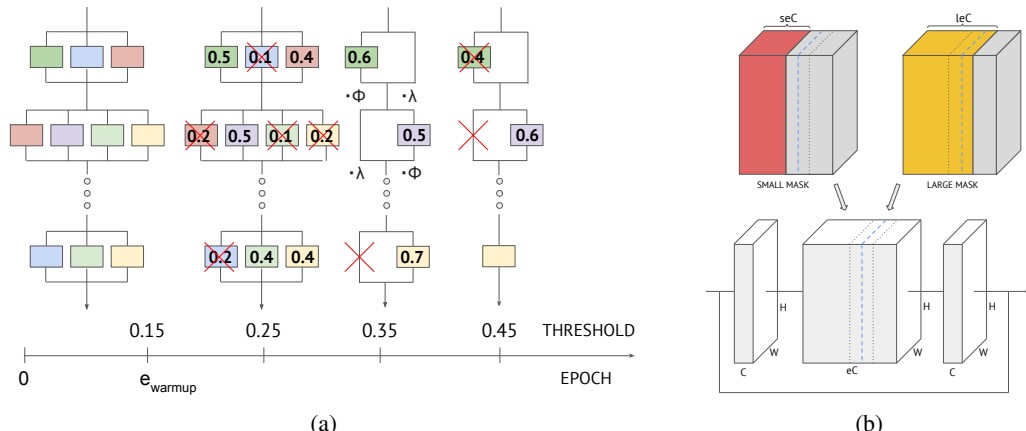

(a)                                                     (b)

Figure 2: (a) **SuperNet trimming procedure**: For the first $e_{warmup}$ epochs, only the regular weights are being trained, and the architecture weights are frozen. After the warmup phase, the threshold increases linearly. If an architecture weight of a block is below the current threshold, it is removed from the layer. If the penultimate is to be removed from the layer, it is replaced by a skip connection block instead. The output of this skip connection block is multiplied by $\phi$, and the output of the other block is multiplied by $\lambda$ as stated in Section 3.2.3. At the end of the training, every layer contains exactly one block; thus, the result of the training is a non-stochastic network.
(b) **Procedure for channel pruning in case of an IRB**: The inner tensor in IRB has the shape of $eC \times W \times H$, where $C$ is the input number of channels of the block, $e$ is the maximal expansion ratio, $W$ and $H$ are the width and the height of the feature map. Small candidate uses $seC$ channels, and large candidate uses $leC$ channels, where $s, l \in (0; 1]$ and $s < l$. The optimal candidate that we try to find (marked with the blue dashed line) uses $oC$ channels, and it is assumed that $s \le o \le l$. Both candidates mask out unused channels. The weights are shared between the candidates. The proportion of channels both candidates use, denoted by $s$ and $l$, dynamically changes throughout the training.

The training is conducted on a smaller proxy dataset to make the SuperNet training faster. We empirically confirmed that the performance of the found architecture on the proxy problem correlates with the performance on the original problem. Once the final architecture is obtained from the SuperNet, it is evaluated and re-trained from scratch on the full dataset.

### 3.2.1 PRUNING INTERNAL STRUCTURE OF A BLOCK

We use a particular procedure for finding the final values of discrete inner hidden dimensions of a block. It is required that the impact of discrete parameters on the objective function (in our case, a function of latency and accuracy) is predictable, e.g., small parameter values mean a negative impact on accuracy but a positive impact on latency, and large parameter values mean a positive impact on accuracy but negative impact on latency. We also require it to be regular – the impacts described above are monotonic with regard to the parameter value. A good example of such a parameter is the expansion ratio in the IRB.

Prunode consists of two copies of the same block. Unlike in Wu et al. (2019), both blocks share the weights Pham et al. (2018), and the only difference between them is masking. In contrast to Wan et al. (2020), masks are not applied to the outputs of blocks but only to a hidden dimension. The larger mask is initialized with all 1s, thus using all the channels. The smaller mask is initialized in a way to mask out half of the channels. Prunodes are benchmarked with all possible inner hidden dimension values. The latency of a prunode is the average of the latencies of the smaller and the larger candidates weighted by Gumbel-Softmax coefficients.

We try to ensure that the optimal mask value $o$ is between the candidates. More precisely, if a candidate with a larger mask $l$ has a higher likelihood of being chosen, i.e., has a larger architecture weight compared to a candidate with a smaller mask $s$, then both masks should be expanded. In the

---

**Algorithm 1:** Prunode masking

Constants $c$, max_distance, granularity, and momentum were set to 0.8, 0.6, 32, and 0.4 respectively.

```
weight ← 0.0;                         // Initialize architecture weight used by Gumbel-Softnax with 0
update ← 0.0;
s ← 0.5;                              // Small mask initially masks out half of the channels
l ← 1.0;                              // Large mask initially contains all of the channels
Procedure update_masks (progress)     // Called after each training iteration
    if (l − s) × max_channels > granularity then  // Update masks until consecutive choices are reached
        update ← update × momentum + weight ;                    // Momentum speeds up convergence
        distance ← max_distance × (1 − progress)²; // Calculate distance between masks; progress ∈ [0, 1]
        s ← s + update;                                          // Update small mask
        if s > 0 then
            weight ← 0;                   // Reset architecture weight if not a corner case
            s ← min(s, 1 − c × distance);                        // Prevent premature convergence
        else
            s ← 0 ;                                              // Make small mask non-negative
        end
        l ← s + distance;                                        // Update large mask
        small_mask ← round(s × max_channels, granularity);       // Ensure divisibility by granularity
        small_mask ← clip(small_mask, granularity, max_channels − granularity); // Ensure small_mask is in bounds
        large_mask ← round(l × max_channels, granularity);       // Ensure divisibility by granularity
        large_mask ← clip(large_mask, small_mask + granularity, max_channels); // Ensure large_mask is in bounds
        create_masks(small_mask, large_mask);
    end
```

---

opposite case, both masks should be reduced. The distance between masks $d = l − s$ should decrease as the training progresses, and at the end of the search, the $d$ should be close to zero. Due to the rules above, by the end of the search, we expect that the final values obtained are close to each other and should tend to an optimized solution. After the search, we sample a single candidate (from those two modified candidates), cf. Wu et al. (2019). Fig. 2b visualizes the masks of two candidates in the case of using IRB. The exact procedure of prunode masking is presented in Algorithm 1.

Our routine extensively searches through discrete inner hidden dimension parameter values while keeping memory usage and computation costs low. Our proposed method has a computation cost and memory usage of $\mathcal{O}(1)$ with respect to the number of all possible values, as only two candidates are evaluated every time. Further, as we prune the not promising candidates, the SuperNet architecture tends to the sampled one. As a result, co-adaptation among operations is minimized.

### 3.2.2 PRUNING BLOCKS WITHIN A STOCHASTIC LAYER

Each block in layer $l$ has its architecture weight initialized as $1/N_l$, where $N_l$ is the number of blocks in the layer $l$. Architecture weights are not modified for the first $e_{\text{warmup}}$ epochs. After the warmup phase is finished, the SuperNet can be pruned, i.e., when a block in a layer has a probability of being chosen below a threshold, it is removed from the layer. This threshold changes linearly during the training and depends only on the value of the current epoch of the training. During epoch $e > e_{\text{warmup}}$ the threshold $t$ is equal to

$$t(e) = t_{\text{initial}} + (t_{\text{final}} − t_{\text{initial}}) \frac{e − e_{\text{warmup}}}{e_{\text{total}} − e_{\text{warmup}}}. \tag{6}$$

The initial threshold should not be higher than $1/\max(N_l)$, where $\max(N_l)$ is the highest number of blocks in a layer, so no blocks are immediately removed after the warmup phase. The final threshold should not be lower than 0.5; thus, all blocks but one are removed from each layer during the training. An example of block pruning is presented in Fig. 2a.

### 3.2.3 PRUNING UNNECESSARY STOCHASTIC LAYERS

Some layers might have the same input and output tensor sizes. For such layers, it is possible to remove them completely to obtain a shorter network. Removing a layer is equivalent to choosing a skip connection block. However, adding a skip connection block to the search space increases memory consumption. Moreover, using this approach, we observed a bias towards finding shallower architectures than optimal. Instead of adding a skip connection block to the set of possible blocks of a layer, we introduce a GPU memory optimization. When the penultimate block is to be removed from

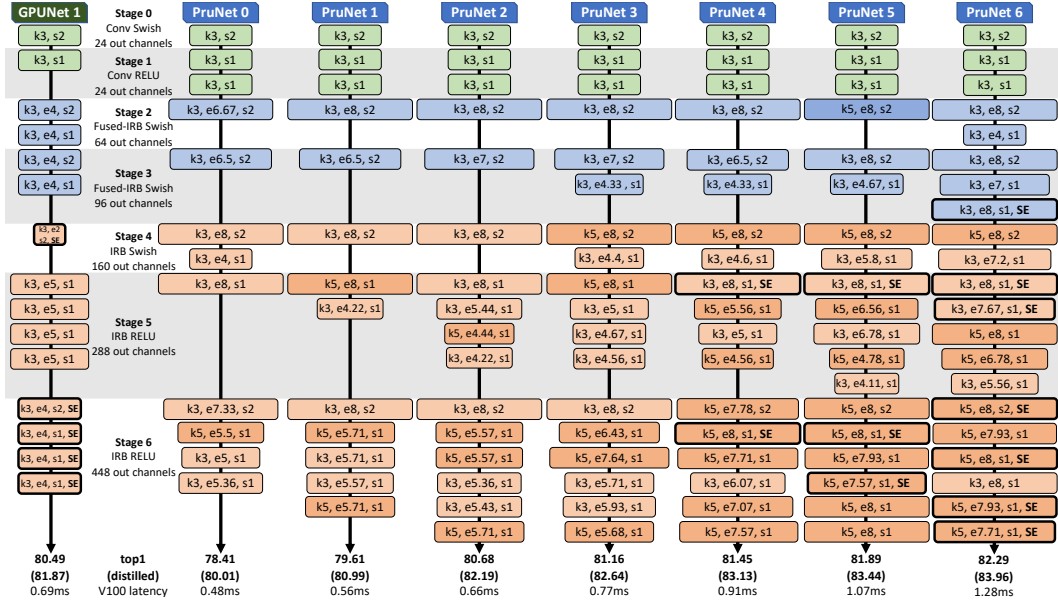

Figure 3: **PRUNET networks**: Image input resolution is set to 288x288 for all networks. Thick border represents block using Squeeze and Excitation. Darker shade of color represents kernel size = 5. Width is proportional to expansion ratio value. Each architecture is followed by Convolution 1x1 with 1280 output channels, RELU activation function, Adaptive Average Pool, and Linear layer.

a layer, it is replaced by a skip connection block. From that point, the output of the skip connection block is multiplied by $\phi$, and the output of the other remaining block is multiplied by the parameter $\lambda$, where $\phi$ and $\lambda$ are coefficients fixed for the whole training. Each layer uses the same values of these parameters. Once the skip connection block or the other block is removed from the layer, the output is no longer multiplied by any parameter. Selecting different $\lambda$ and $\phi$ values allows to reduce the bias towards shallower networks and thus gives more control to find an even better architecture.

## 4 EXPERIMENTS

We test our methodology on Imagenet-1k Deng et al. (2009) as there are many good networks that can potentially be tuned. In particular, recent GPUNet networks Wang et al. (2022) set up a SOTA Pareto frontier in terms of latency and inference accuracy for small networks. Their architecture scheme is perfect to showcase the value of our proposed method. The networks created during these experiments are called PRUNET, and Fig. 3 presents their structure. In this section, we delve into details of how we found PRUNET family and analyze the search cost. We also show that our networks transfer well to COCO object detection task.

### 4.1 IMAGE CLASSIFICATION ON IMAGENET

#### 4.1.1 FINDING PRUNET NETWORK FAMILY

Inspired by GPUNet-1 architecture, we define our SuperNet, as described in section 3.1 and Table 1, with an image resolution of 288x288, 2D convolution with kernel size of 3, stride of 2, and Swish Ramachandran et al. (2017) activation function as the prologue, and then define 6 stages followed by an epilogue. At each stage, we use the same type of building block (convolution, IRB, or Fused Inverted Residual Block denoted by Fused-IRB), activation function, and a number of channels as in GPUNet-1. Within these constraints in stages from 2 to 6, we define stochastic layers that consist of four multi-blocks (kernel size $\in \{3,5\}$, SE $\in \{$True, False$\}$), all with maximum expansion of 8 and granularity of 32 channels. SE stands for Squeeze and Excitation block, cf. Hu et al. (2018). For stage 1, we define only two choices – kernel size $\in \{3,5\}$. Each stage contains one additional layer

that can be skipped during the training compared to GPUNet-1. Thanks to such a defined SuperNet, we can sample GPUNet-1; hence the final result is expected to be improved.

To generate the entire Pareto frontier, we experiment with 7 different values of $\alpha \in \{0.2, 0.4, 0.6, 0.8, 1.0, 1.2, 2.0\}$ . Parameter $\alpha$ is defined in equation 2 and changes the trade-off between latency and accuracy. Since proper layer skipping is crucial for finding a good network, and the last 130 epochs are relatively inexpensive due to the progressive pruning of the search space, we test 3 variants of layer skipping for each $\alpha$. On the basis of preliminary experiments, we chose $\lambda \in \{0.4, 0.55, 0.85\}$ and $\phi = 1.1$. For each $\alpha$ value, we decided which $\lambda$ value was the best based on the final loss from the SuperNet search. The architecture with the best loss was then trained from scratch. Table 2 compares the PRUNET results against other baselines. For all the considered networks of comparable accuracy, PRUNET has significantly lower latency. It is worth noting that it does not necessarily have a lower number of parameters or FLOPs. In particular, comparing PRUNETs and GPUNets (both among distilled and non-distilled networks), we observe that the obtained networks are uniformly better, meaning we get higher accuracy with lower latency.

### 4.1.2 ABLATION STUDY

Each of the optimizations has a different impact on search performance. Main advantage of the first one (3.2.1) is memory efficiency. Prunode allows significant increase of the size of the search space without increasing the demand for compute and memory resources. The second optimization (3.2.2) can save a lot of computational time without compromising the quality of the results. Last optimization (3.2.3) saves memory that would be needed for skip connection in standard FBNet approach and enhances the quality of the end results by reducing the impact of curse of skip connect. Table 3 visualizes the cost impact of each optimization in a hard memory limit scenario. Appendix B further investigates the impact on the quality of the results.

### 4.2 OBJECT DETECTION ON COCO

The experiments were conducted on the COCO 2017 detection datasetLin et al. (2014). We chose EfficientDet Tan et al. (2020) as a baseline model. We replaced the original EfficientNet backbone with GPUNet and PRUNET for broad comparison. All backbones were pretrained without distillation. Figure 4 shows that our architectures can be successfully transferred to other Computer Vision tasks. PRUNET turned out to be faster and more accurate, similarly as it was on the image classification task.

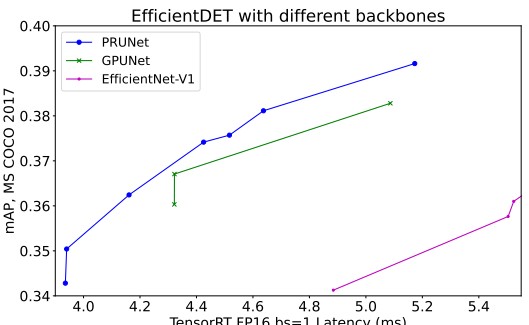

Figure 4: PRUNET as a backbone outperforms GPUNet and EfficientNet on COCO object detection task on inference latency relative to mean Average Precision (mAP)

## 5 CONCLUSIONS

We introduced **Prunode** a stochastic bi-path building block that can be used to search the inner hidden dimension of blocks in any differentiable NAS with $\mathcal{O}(1)$ cost. Together with two novel layer pruning techniques, we show that our proposed inference-aware method establishes a new SOTA Pareto frontier (PRUNET) in TensorRT inference latency and ImageNet-1K top-1 accuracy and enables fine granularity sampling of the Pareto frontier to better fit external deployment constraints. Further, our results in Table 2 confirm that FLOP and the number of parameters are not the suitable proxies for latency, highlighting the importance of the correct metric for architectural evaluation. Although we only present results within computer vision tasks, our methods can be generalized to searching models for natural language processing (NLP), speech, and recommendation system tasks.

Table 2: PRUNET image classification results. All latency measurements are made using batch size 1.

| Models | Top1 ImageNet | TensorRT Latency FP16 V100 (ms) | #Params (10e6) | #FLOPS (10e9) | PRUNET Speedup ↑ | PRUNET Accuracy ↑ |
|---|---|---|---|---|---|---|
| PRUNET Without Distillation Comparison | | | | | | |
| EfficientNet-B0 Tan and Le (2019) | 77.1 | 0.94 | 5.28 | 0.38 | 1.96x | 1.3 |
| EfficientNetX-B0-GPU Li et al. (2021) | 77.3 | 0.96 | 7.6 | 0.91 | 2x | 1.1 |
| REGNETY-1.6GF Radosavovic et al. (2020) | 78 | 2.55 | 11.2 | 1.6 | 5.31x | 0.4 |
| **PRUNET -0** | 78.4 | 0.48 | 11.3 | 2.10 | | |
| OFA 389 Cai et al. (2019) | 79.1 | 0.94 | 8.4 | 0.39 | 1.68x | 0.5 |
| EfficientNetX-B1-GPU | 79.4 | 1.37 | 9.6 | 1.58 | 2.45x | 0.2 |
| OFA 482 | 79.6 | 1.06 | 9.1 | 0.48 | 1.89x | 0.0 |
| **PRUNET -1** | 79.6 | 0.56 | 14.8 | 2.57 | | |
| FBNetV3-B Dai et al. (2020) | 79.8 | 1.5 | 8.6. | 0.46 | 2.27x | 0.9 |
| EfficientNetX-B2-GPU | 80.0 | 1.46 | 10 | 2.3 | 2.21x | 0.7 |
| OFA 595 | 80.0 | 1.09 | 9.1 | 0.6 | 1.65x | 0.7 |
| EfficientNet-B2 | 80.3 | 1.65 | 9.2 | 1 | 2.5x | 0.4 |
| ResNet-50 He et al. (2016) | 80.3 | 1.1 | 28.09 | 4. | 1.67x | 0.4 |
| GPUNet-1 Wang et al. (2022) | 80.5 | 0.69 | 12.7 | 3.3 | 1.04x | 0.2 |
| **PRUNET -2** | 80.7 | 0.66 | 18.6 | 3.31 | | |
| REGNETY-32GF | 81 | 4.97 | 145 | 32.3 | 6.45x | 0.2 |
| **PRUNET -3** | 81.2 | 0.77 | 20.8 | 4.05 | | |
| EfficientNetX-B3-GPU | 81.4 | 1.9 | 13.3 | 4.3 | 2.09x | 0.1 |
| **PRUNET -4** | 81.5 | 0.91 | 24.0 | 4.25 | | |
| EfficientNet-B3 | 81.6 | 2.04 | 12 | 1.8 | 1.91x | 0.3 |
| **PRUNET -5** | 81.9 | 1.07 | 27.5 | 5.29 | | |
| ResNet-101 He et al. (2016) | 82.0 | 1.7 | 45 | 7.6 | 1.33x | 0.3 |
| FBNetV3-F | 82.1 | 1.97 | 13.9 | 1.18 | 1.54x | 0.2 |
| GPUNet-2 Wang et al. (2022) | 82.2 | 1.57 | 25.8 | 8.38 | 1.23x | 0.1 |
| **PRUNET -6** | 82.3 | 1.28 | 31.1 | 7.36 | | |
| PRUNET With Distillation Comparison | | | | | | |
| **PRUNET -0 (distilled)** | 80.0 | 0.48 | 11.3 | 2.10 | | |
| GPUNet-0 (distilled) | 80.7 | 0.61 | 11.9 | 3.25 | 1.09x | 0.3 |
| **PRUNET -1 (distilled)** | 81.0 | 0.56 | 14.8 | 2.57 | | |
| GPUNet-1 (distilled) | 81.9 | 0.69 | 12.7 | 3.3 | 1.04x | 0.3 |
| **PRUNET -2 (distilled)** | 82.2 | 0.66 | 18.5 | 3.31 | | |
| **PRUNET -3 (distilled)** | 82.6 | 0.77 | 20.8 | 4.05 | | |
| **PRUNET -4 (distilled)** | 83.1 | 0.91 | 24.0 | 4.25 | | |
| **PRUNET -5 (distilled)** | 83.4 | 1.07 | 27.5 | 5.29 | | |
| GPUNet-2 (distilled) | 83.5 | 1.57 | 25.8 | 8.38 | 1.23x | 0.5 |
| EfficientNetV2-S Tan and Le (2021) | 83.9 | 2.67 | 22 | 8.8 | 2.09x | 0.1 |
| **PRUNET -6 (distilled)** | 84.0 | 1.28 | 31.1 | 7.36 | | |

Table 3: **Cost analysis in the 80 GB memory limit scenario**. Train time considers searching for all PRUNET architectures on a node with 8 NVIDIA Tesla A100 GPUs (DGX-A100). Section 3.1 contains the calculations of the search space sizes.

| | FBNet | 1st opt (3.2.1) | 2nd opt (3.2.2) | 3rd opt (3.2.3) | All opts |
|---|---|---|---|---|---|
| Memory consumption [GB] | 80 | 80 | 80 | 80 | 80 |
| Search space size | 9e15 | 7.1e38 | 9e15 | 1.5e17 | 1.7e39 |
| Train time [h] | 140 | 140 | 62 | 140 | 62 |

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

## A    EXPERIMENTS DETAILS

### A.1    MACHINE AND SOFTWARE SETUP

The experiments were performed on nodes equipped with 8 NVIDIA Tesla A100 GPUs (DGX-A100). Python 3.6.8 and Pytorch Image Models Wightman (2019) v0.4.12 were used inside `pytorch/pytorch:1.9.1-cuda11.1-cudnn8-runtime` docker container.

By inference latency we mean *Median GPU Compute Time* measured via `trtexec` command with FP16 precision and batch size of 1 using TensorRT version 8.0.1.6 on a single NVIDIA V100 GPU with driver version 450.51.06 inside `nvcr.io/nvidia/tensorrt:21.08-py3` docker container.

### A.2    IMAGENET EXPERIMENTS

The experiments were conducted on the Imagenet-1k Deng et al. (2009) image classification dataset. It consists of 1.2 million training samples and 50 thousand validation samples, which span over 1000 classes. For all experiments we used a weight decay of 1e-5 and an AutoAugment Cubuk et al. (2018) with an augmentation magnitude of 9 and standard deviation 0.5 (corresponding to the probability of applying the operation) for both architecture search and training from scratch. The experiments were performed in automatic mixed precision (AMP).

#### A.2.1    ARCHITECTURE SEARCH DETAILS

An architecture search was performed on $10\%$ of randomly selected classes from the original dataset. The input images were scaled to a resolution of $288 \times 288$. The search lasted 200 epochs, with a total batch size of 256 and a cosine learning rate scheduler that had an initial value of 0.1. We used the Adam Kingma and Ba (2014) optimizer for the architecture parameters and the RMSprop optimizer with an initial learning rate of 0.002 for the weights. We divided the search into two phases. In the first phase we train only the regular weights for the first $e_{\text{warmup}} = 70$ epochs, and we only do it once to save computational time. In the second phase, which lasts the remaining 130 epochs, in each epoch $80\%$ of the training dataset was used to train the regular weights and the remaining $20\%$ was used to train the architecture weights. The second phase has been computed in many variants but always starts from a common checkpoint after $e_{\text{warmup}}$ epochs. During this phase, we progressively prune the search space using a pruning threshold (see Section 3.2.2) that increases linearly from 0.15 to 0.55. At the end of each epoch, we removed blocks below the threshold from the search space. The momentum used in the pruning internal structure of a Prunode was equal to 0.4 . The coefficient $\alpha$ in the loss function varied across different runs with analyzed values of $\{2.0, 1.2, 1.0, 0.8, 0.6, 0.4, 0.2\}$, but all runs used the same coefficient $\beta$ value of 0.6 . The latency term LAT of the loss function was measured in µs.

#### A.2.2    TRAINING DETAILS

The hyperparameters' values were based on GPUNet-1 Wang et al. (2022). For fair comparison we decided to train GPUNet-0, GPUNet-1 and GPUNet-2 using exactly the same hyperparameters (including batch size) as we used for PRUNETs .

After the architecture search, the sampled network was again trained from scratch. The training lasted for 450 epochs with a total batch size of 1536 and an initial learning rate of 0.06. The learning rate decays by 0.97 times for every 2.4 epochs. `crop_pct` was set to 1.0. Exponential Moving Average (EMA) was used with a decay factor of 0.9999. We used the drop path Larsson et al. (2016) with a base drop path rate of 0.2. All of the PRUNET and GPUNet networks have been trained with and without distillation Hinton et al. (2015). Knowledge distillation is a technique that transfers the knowledge from a large pre-trained model to a smaller one which can be deployed under real-world limited constrains. For the training with distillation, we used different teachers and different `crop_pct` as it is presented in Table 4.

Table 4: For each image resolution a different teacher was selected. Additionally `crop_pct` was changed to match the `crop_pct` of the teacher.

| | model resolution | teacher architecture | teacher resolution | `crop_pct` |
|---|---|---|---|---|
| GPUNet1 & PRUNET | $288 \times 288$ | EfficientNet-B3 | $300 \times 300$ | 0.904 |
| GPUNet0 | $320 \times 320$ | EfficientNet-B4 | $380 \times 380$ | 0.922 |
| GPUNet2 | $384 \times 384$ | EfficientNet-B5 | $456 \times 456$ | 0.934 |

Table 5: **Search space for B.1 experiment**:
For stages 2-6 the search space is defined as the sum of RELU and Swish variants

| Stage | Type | Stride | # Layers | Activation | Kernel | Expansion | Filters |
|---|---|---|---|---|---|---|---|
| 0 | Conv | 1 | 1 | Swish | 3 | | 24 |
| 1 | Conv | 1 | 2 | RELU | {3,5,7} | | 24 |
| 2 | Fused-IRB | 2 | 2 | RELU | {3,5} | 8 | 64 |
| | | | | Swish | {3,5,7} | | |
| 3 | Fused-IRB | 2 | 3 | RELU | {3,5} | 8 | 96 |
| | | | | Swish | {3,5,7} | | |
| 4 | IRB | 2 | 2 | RELU | {3,5} | 8 | 160 |
| | | | | Swish | {3,5,7} | | |
| 5 | IRB | 1 | 5 | RELU | {3,5,7} | 8 | 288 |
| | | | | Swish | {3,5} | | |
| 6 | IRB | 2 | 6 | RELU | {3,5,7} | 8 | 448 |
| | | | | Swish | {3,5} | | |
| 7 | Conv+Pool+FC | 1 | 1 | RELU | 1 | | 1280 |

## A.3 COCO EXPERIMENTS

The object detection experiments were conducted on the MS COCO 2017 dataset Lin et al. (2014). We used EfficientDet Tan et al. (2020) as a baseline model and replaced the original EfficientNet Tan and Le (2019) backbone with PRUNET and GPUNet. The training lasted for 300 epochs with batch size of 60. The learning rate was warmed-up for the first 20 epochs with the value set to 1e-4. Then, the cosine learning rate scheduler was used with an initial learning rate of 0.65. The optimizer was SGD with a momentum of 0.9 and a weight decay of 4e-5. Gradient clipping of value 10.0 was introduced. The training was performed in automatic mixed precision (AMP). We used the Exponential Moving Average (EMA) with a decay factor of 0.999.

## B ADDITIONAL EXPERIMENTS

### B.1 PRUNING BLOCKS WITHIN A STOCHASTIC LAYER

We can have two different versions of setting the pruning threshold, i.e., constant or linearly increasing. The hypothesis was that they lead to the networks of comparable Pareto frontiers, but the search is significantly faster for the linearly increasing threshold. The experiments supported this hypothesis, and the result Pareto frontiers are on par; however, the processing time for the constant threshold was around $15\% - 45\%$ slower.

We used the Imagewoof Howard image classification dataset both for the search and the training of the obtained architectures. The search space is described in Table 5.

The search was performed with batch size 256, cosine learning rate scheduler with initial value of 0.1. It lasted 200 epochs and for the first $e_{\text{warmup}} = 70$ epochs only the regular weights were trained. The image resolution was scaled to $224 \times 224$ and `crop_pct` was set to 0.875. Adam optimizer was used for architecture parameters and RMSprop optimizer with initial learning rate of 0.002 for the regular weights. A weight decay of 1e-5 was used and AutoAugment with an augmentation magniture of 9 and standard deviation of 0.5. The search was performed in automatic mixed precision (AMP). In each epoch $80\%$ of the training dataset was used to train the regular weights and the reamaining

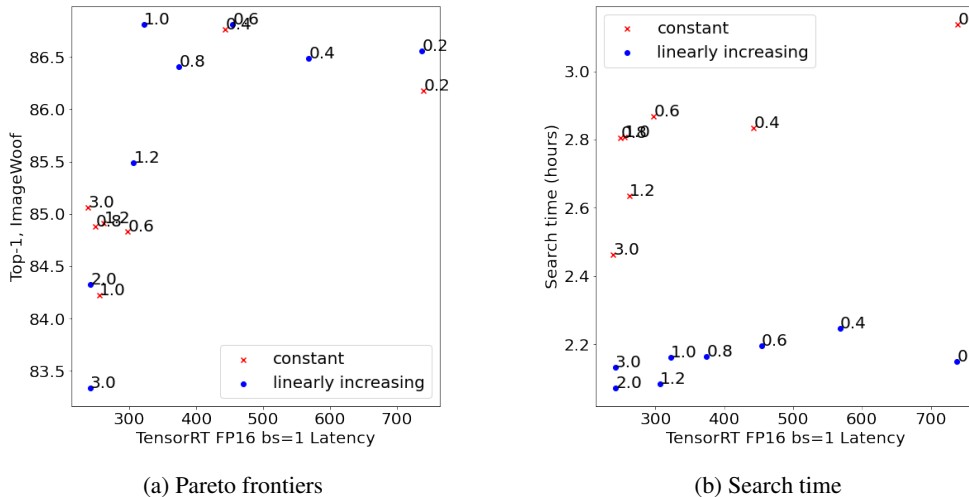

(a) Pareto frontiers  (b) Search time

Figure 5: Linearly increasing pruning threshold leads to a significant search phase speedup while producing a similar Pareto frontier to a constant threshold method. Labels indicate the value of $\alpha$ being used.

$20\%$ was used to train the architecture weights. For a constant pruning threshold policy, we use a value of $0.133$; for the second setup, we linearly increase the threshold from $0.133$ to $0.55$ (see Section 3.2.2). The coefficient $\alpha$ in the loss function varied across different runs with analyzed values of $\{0.2, 0.4, 0.6, 0.8, 1.0, 1.2, 2.0, 3.0\}$, but all runs used the same coefficient $\beta$ value of $0.6$.

After the search has finished, the architectures have been re-trained from scratch for $450$ epochs in automatic mixed precision (AMP) using Stochastic Gradient Descent (SGD) with batch size of $32$, and initial learning rate of $0.2$. The learning rate decays by $0.97$ for every $2.4$ epochs. The drop path with bsae drop rate of $0.2$ was used. The results of those experiments are presented in Figure 5.

## B.2 PRUNING UNNECESSARY STOCHASTIC LAYERS

Proper selection of parameters $\phi$ and $\lambda$ (which multiply the output of skip connection and the output of a block; for more details, see Section 3.2.3) can influence the final length of the sampled network. In particular Table 6 shows that for $\alpha$ large enough, the number of layers is inversely related to $\lambda$ for considered cases. To further analyze the impact of our methodology, we run additional searches with $(\phi, \lambda) = (1.0, 1.0)$ – these values correspond to a base approach without our modification. From Table 6 it is clear that for $\alpha \geq 0.8$, the base approach samples networks with fewer layers than the searches we performed. To prove that our approach is justified, we train all 28 of the architectures found during the searches described above, and we present Pareto frontiers defined by all four sets of $\phi$ and $\lambda$. Figure 6 shows that our search technique can find networks with up to $15\%$ better latency with the same accuracy as the base approach. Our methodology uses the final loss of the search as a zero-cost filter to select good candidate architectures to be evaluated. Figure 7 presents the Pareto frontier of Search Loss concerning Final Latency. Looking at both figures, it is directly visible that if search loss is significantly better for similar searches (in this case for similar $\alpha$ values), we can also expect better results in terms of the final accuracy; however, the correlation is not strong. Therefore, without additional compute overhead, our methodology can find almost optimal networks, possibly still leaving room for improvement.

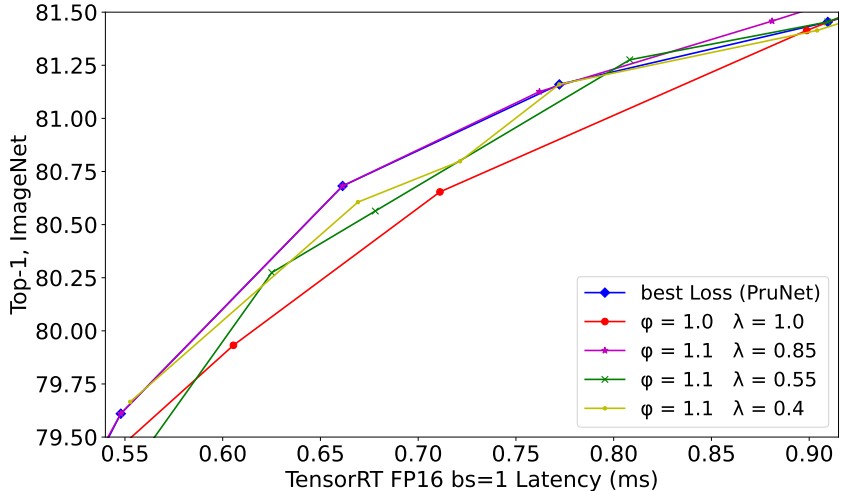

Figure 6: Graph focuses on the architectures being searched using $0.6 \leq \alpha \leq 1.2$

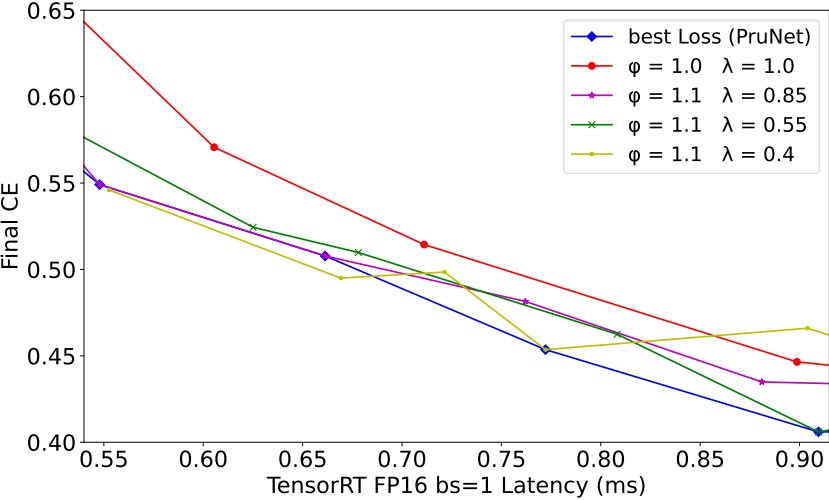

Figure 7: Graph focuses on the architectures being searched using $0.6 \leq \alpha \leq 1.2$, which shows the relationship between the final CE of the search and the final latency of the architecture

Table 6: Search results for different $\phi$ and $\lambda$.

For each combination of $(\alpha, \phi, \lambda)$ we run a single search and for each search the total loss is reported. The total loss is a sum of cross entropy loss and latency loss (Loss = CE + $\alpha * \text{LAT}^{\beta}$). #IRBs indicates the number of Fused-IRB and IRB in the final architecture. The number of the other layers is the same for all architectures. PRUNET architectures are shown in bold

|  | $\phi, \lambda = (1.0, 1.0)$ | | $\phi, \lambda = (1.1, 0.85)$ | | $\phi, \lambda = (1.1, 0.55)$ | | $\phi, \lambda = (1.1, 0.4)$ | |
|---|---|---|---|---|---|---|---|---|
|  | Loss | #IRBs | Loss | #IRBs | Loss | #IRBs | Loss | #IRBs |
| $\alpha = 2.0$ | 6.7149 | 6 | 6.6316 | 7 | **6.5968** | **9** | 6.6241 | 11 |
| $\alpha = 1.2$ | 4.2938 | 9 | **4.1876** | **10** | 4.2139 | 12 | 4.2039 | 13 |
| $\alpha = 1.0$ | 3.6352 | 11 | **3.6015** | **13** | 3.6074 | 13 | 3.6138 | 14 |
| $\alpha = 0.8$ | 3.0027 | 13 | 2.9932 | 14 | 2.9762 | 14 | **2.9692** | **15** |
| $\alpha = 0.6$ | 2.3528 | 15 | 2.3394 | 15 | **2.316** | **15** | 2.3723 | 15 |
| $\alpha = 0.4$ | 1.7154 | 16 | 1.7201 | 17 | 1.7296 | 16 | **1.6946** | **16** |
| $\alpha = 0.2$ | 1.0573 | 18 | 1.0573 | 18 | 1.0687 | 18 | **1.0531** | **18** |

