# OpenReview forum: "Tiered Pruning for Efficient Differentialble Inference-Aware Neural Architecture Search"
_ICLR.cc/2023/Conference — Submitted to ICLR 2023_

### Official Review · Reviewer_Xpgv · 2022-10-24

**Confidence:** 5
**Correctness:** 2
**Technical Novelty And Significance:** 2
**Empirical Novelty And Significance:** 2
**Recommendation:** 3

**Clarity, Quality, Novelty And Reproducibility:**

- The method is easy to understand, but the experiment details are not clear.
- The experiment results are quite good, but this does not mean that the experiment quality of this paper is high. I believe most of the comparisons are not fair in this paper.
- Novelty is fair.
- Reproducibility is not guaranteed since the authors did not release the code and did not claim that code will be released.

**Strength And Weaknesses:**

Strengths:
- The authors carry out very extensive experiments on ImageNet and COCO, achieving impressive accuracy results.
- The proposed method itself is easy to understand.

Weaknesses:
- The writing of this paper needs to be improved. It seems to me that this paper is a tech report rather than a research paper. For citations in the paper, please make sure that the whole sentence can read. For example,

> recent research has focused on partial training Falkner et al. (2018); Li et al. (2020a); Luo et al. (2018), performing network morphism ...

in p1 does not seem to be a readable sentence. The authors should also consider to replace $*$ with $\times$ in equation in the end of page 3.

- Despite good results, the method itself has very limited novelty. Modeling the NAS problem as bi-level optimization, using Gumble softmax to convert non-differentiable HPO problems to differentiable optimization problems, and adding a latency term in the loss function are all very well-recognized techniques in the NAS community. The design space used in this paper is also highly inspired by EfficientNetV2. One can also easily find the idea of gradually shrinking the NAS design space in [AtomNAS](https://arxiv.org/pdf/1912.09640.pdf) of ICLR'20.
- The comparison in Figure 4 is not fair. The method is clearly inspired by EfficientNetV2, but it compares itself with an apparently weaker baseline, EfficientDet (backed by EfficientNetV1).
- The authors never made a head-to-head comparison against one-shot NAS (e.g. Single Path One Shot NAS in ECCV'20) or other differentiable NAS methods in the same search space, which makes it very hard for me to judge the merit of this paper against existing approaches.

**Summary Of The Paper:**

In this paper, the authors proposed a differentiable NAS approach that searches for efficient CNN models on GPUs. The key idea is similar to existing differentiable NAS methods: modeling NAS as a bi-level optimization problem where both weight and architectural parameters are updated during SuperNet training. The paper proposes to dynamically shrink the search space during the procedure of super network training. After super network training, operators with the largest architectural parameter is kept for each searchable block, which is also the standard procedure in differentiable NAS.

**Summary Of The Review:**

Despite good results, this paper fails to make fair comparisons with existing methods and has very limited novelty. Without a major reformat, I cannot see the possibility of recommending acceptance.

---

### Official Review · Reviewer_GWQr · 2022-10-25

**Confidence:** 3
**Correctness:** 3
**Technical Novelty And Significance:** 2
**Empirical Novelty And Significance:** 3
**Recommendation:** 5

**Clarity, Quality, Novelty And Reproducibility:**

- Clarity: The paper is a bit hard to read. The naming of variables seems to be inconsistent, and to some extent, causal (e.g., those in Algo 1).

- Quality and Novelty: please see the above.

- Reproducibility: It can be hard to reproduce as many hyperparameters exist, which may require domain-expertise for careful tuning.

**Strength And Weaknesses:**

Strengths:

- The authors propose to prune the search space for efficient neural architecture search. The pruning is conducted on different network angularities: inner block level, inter block level and layer level.

- Experimental results show that PRUNET achieves new SOTA w.r.t the inference latency and classification accuracy on ImageNet.

- The authors also provide thorough ablation studies over the hyperparameters of the algorithm.


Weakness:

- A major concern with this paper is the heuristics that make the procedure overcomplicated. There are also a number of associated hyper-parameters, e.g., constant $c$, max_distance, momentum in Section 3.2.1, $e_{\textrm{warmup}}$, $e_{\textrm{warmup}}$, $t_{\textrm{initial}}$ and $t_{\textrm{final}}$ in Section 3.2.2, and $\lambda$, $\phi$ in Section 3.2.3. Although authors provide some ablations, these hyperparameters make the approach hard to tune in practice.

- Algo 1 is hard to follow. More explanations can be provided.  It is also not clear to me in what way can the task loss affect the mask training, since their updates are based on heuristic rules.

- It seems not new to have O(1) memory and computation complexity. It is common for popular NAS methods (e.g., ProxylessNAS [1]) to sample one path from the cell that achieves the same complexity.

Detailed comments:

- How do you construct the latency table? Are the latency values measured based on TensorRT conversion? Besides, for baselines such as EfficientNet, do you apply the same TensorRT conversion before measuring their latencies?

- It seems the algorithm cannot regularize the model size explicitly. Thus picking different sizes of PRUNET architectures from Table 6 can be tricky.

- "small parameter values mean a negative impact on accuracy but a positive impact on latency, and large parameter values mean a positive impact on accuracy but negative impact on latency": what do you refer by "small parameter value"? Is it the architecture weight or the network parameter?


[1] Cai H, Zhu L, Han S. Proxylessnas: Direct neural architecture search on target task and hardware[J]. arXiv preprint arXiv:1812.00332, 2018.

**Summary Of The Paper:**

This paper studies a new approach with the purpose of improving the efficiency of differentiable neural architecture search (DNAS). The authors introduce Prunode, a stochastic bi-path building block that enjoys O(1) memory and computation complexity during the search. Given this advantage, Prunode allows a much larger search space than conventional NAS algorithms. To further reduce the searching computation, the authors also propose to prune the blocks within the stochastic layer that are less likely to be chosen, as well as unnecessary layers. Experimental results show that PRUNET establishes the new state-of-the-art Pareto frontier w.r.t latency and accuracy on ImageNet.

**Summary Of The Review:**

The paper studies the important problem of efficient neural architecture search, and experimental results are thorough and solid. I can see the efforts from authors to improve searching efficiency by various practical designs. Yet, my major concern is that too many heuristics involved in the proposed approach that make it hard to implement in practice. The presentation can also be improved.

---

### Official Review · Reviewer_qypu · 2022-10-26

**Confidence:** 5
**Clarity, Quality, Novelty And Reproducibility:** 1. The writing of this paper needs to…
**Correctness:** 3
**Technical Novelty And Significance:** 2
**Empirical Novelty And Significance:** 2
**Recommendation:** 3

**Strength And Weaknesses:**

Strength:

1. the idea in this paper is pretty straightforward
2. the results look promising but not necessarily a solid improvement.


**Summary Of The Paper:**

The paper expands the current supernet training by introducing 3 components:

1. Prunenode: although the name is a bit fancy, the concept is straightforward: it learns the expansion ratio of inverted residual blocks.
2. Pruning blocks with stochastic layers: it kicks out the low likelihood blocks from the search space.
3. Pruning unnecessary stochastic layers: this tries to search over the number of supernet layers that skips certain layers.


**Summary Of The Review:**

1. novelty: This is my main concern: a. The three claimed new components: i) searching for extension layers, ii) searching for blocks in an IRB, iii) search for the number of layers in a network are all previously discussed in Once-For-All[1].

2. The curation of search space borrows from GPUNet. It will help a lot if the authors can clarify the contribution of this paper to the scientific community.

[1] Cai, Han, et al. "Once-for-all: Train one network and specialize it for efficient deployment." arXiv preprint arXiv:1908.09791 (2019).

3. What's the performance if without starting from a good network?

---

### Official Review · Reviewer_Jf9e · 2022-10-27

**Confidence:** 4
**Correctness:** 3
**Technical Novelty And Significance:** 2
**Empirical Novelty And Significance:** 2
**Recommendation:** 3

**Clarity, Quality, Novelty And Reproducibility:**

The paper can be written more clearly. They described the setup clearly so the experiments can be reproduced. The novelty of the work is limited.

**Strength And Weaknesses:**

Strength:
1. They are able to find networks with higher accuracy and lower latency than all their baselines for both image classification and object detection.
2. They use a novel search space - GPUNet which is better suited for building architectures with high accuracy and low latency.

Weakness/ Questions:
 1. Are we using masking for searching for kernels too?
 2. The algorithm is not very straightforward to understand.
3. How is this masking algorithm better than the masking technique used in FBNet V2? If we use the masking technique of FBNet V2 with the same search space and setup as this paper, how would it fare? FBNet V2 requires only 1 block per layer while searching for channels. So it seems to be occupying lesser memory.

Questions about algorithm 1
1. What does progress variable denote? Is it the ratio of number of epochs done to number of epochs left?
2. Why are we setting the architecture weights to 0 if s > 0?
3. f weights are always non-negative, then update variable will also be non-negative. When will s become negative?



**Summary Of The Paper:**

This paper proposes a multi-objective neural architecture search that optimizer for both accuracy and latency. They use GPU-Net1 as the skeleton and search for kernel size and the expansion ratio of the intermediate Fused-IRB and IRB layers. They use masked differentiable architecture search, where the output of a layer is the weighted sum of all the outputs of the blocks. The weights of these blocks are computed using gumbel-softmax of the architecture weights of all the blocks in layer i.
   The input and output dimensions of all the blocks in a layer are the equal. To search for the expansion ratio, rather than have 1 block for each possible configuration, they use masks similar to FBNetV2. To this end, they create two copies of a block and apply small mask on one and a large mask on another. The small mask masks half of the channels to begin with and the larger mask does not mask any of the channels.  As the search progress, the small mask and the larger mask are updated such that the distance between them reduces and finally becomes close to 0. If the architecture weights of the block with small mask is higher, then both the masks shrink. Similarly, if the architecture weights of the block with the larger mask is higher, then both the masks are increased. Finally when the small mask and the larger mask are less than granularity away, the final candidate has channels between the smaller and the larger one.
   During the search, they also prune the blocks if their weights are lesser than the threshold. As is the case with other pruning based NAS methods, the networks weights are trained for the first X epochs. After that the bilevel optimization of network weights and architecture weights begins. The threshold keeps increasing as the search progresses.

**Summary Of The Review:**

The algorithm1 is not clearly written. It is not evident why this is better than the masking technique in FBNet V2.

---

### Decision · Program_Chairs · 2023-01-20

**Decision:**

Reject

**Justification For Why Not Higher Score:**

Everyone agrees to reject.

**Justification For Why Not Lower Score:**

N/A

**Metareview: Summary, Strengths And Weaknesses:**

All reviewers gave rejection scores and the authors did not rebut. Therefore, I follow the reviewers to recommend rejection.